# Coevolution-based inference of amino acid interactions underlying protein function

**Victor H Salinas[1], Rama Ranganathan[2,3]\***

[1]Green Center for Systems Biology, UT Southwestern Medical Center, Dallas, United States; [2]Center for Physics of Evolving Systems, Biochemistry and Molecular Biology, The University of Chicago, Chicago, United States; [3]Institute for Molecular Engineering, The University of Chicago, Chicago, United States

**Abstract** Protein function arises from a poorly understood pattern of energetic interactions between amino acid residues. Sequence-based strategies for deducing this pattern have been proposed, but lack of benchmark data has limited experimental verification. Here, we extend deep-mutation technologies to enable measurement of many thousands of pairwise amino acid couplings in several homologs of a protein family – a deep coupling scan (DCS). The data show that cooperative interactions between residues are loaded in a sparse, evolutionarily conserved, spatially contiguous network of amino acids. The pattern of amino acid coupling is quantitatively captured in the coevolution of amino acid positions, especially as indicated by the statistical coupling analysis (SCA), providing experimental confirmation of the key tenets of this method. This work exposes the collective nature of physical constraints on protein function and clarifies its link with sequence analysis, enabling a general practical approach for understanding the structural basis for protein function.
DOI: https://doi.org/10.7554/eLife.34300.001

**\*For correspondence:**
ranganathanr@uchicago.edu

**Competing interests:** The authors declare that no competing interests exist.

## Introduction

The basic biological properties of proteins – structure, function, and evolvability – arise from the pattern of energetic interactions between amino acid residues (*Anfinsen, 1973*; *Gregoret and Sauer, 1993*; *Luque et al., 2002*; *Starr and Thornton, 2016*; *Weinreich et al., 2006*). This pattern represents the foundation for defining how proteins work, for engineering new activities, and for understanding their origin through the process of evolution. However, the problem of deducing this pattern is extraordinarily difficult. Amino acids act heterogeneously and cooperatively in contributing to protein fitness, properties that are not simple, intuitive functions of the positions of atoms in atomic structures (*Alber et al., 1987*). Indeed, the marginal stability of proteins and the subtlety of the fundamental forces make it so that many degenerate patterns of energetic interactions could be consistent with observed protein structures. The lack of knowledge of this pattern has precluded effective mechanistic models for the relationship between protein structure and function.

In principle, an experimental approach for deducing the pattern of interactions between amino acid residues is the thermodynamic double mutant cycle (*Carter et al., 1984*; *Hidalgo and MacKinnon, 1995*; *Horovitz and Fersht, 1990*) (TDMC, *Figure 1A*). In this method, the energetic coupling between two residues in a protein is probed by studying the effect of mutations at those positions, both singly and in combination. The idea is that if mutations $x$ and $y$ at positions $i$ and $j$, respectively, act independently, the effect of the double mutation ($\Delta G_{ij}^{xy}$) must be the sum of the effects of each single mutant ($\Delta G_i^x + \Delta G_j^y$). Thus, one can compute a coupling free energy between the two mutations ($\Delta\Delta G_{ij}^{xy}$) as:

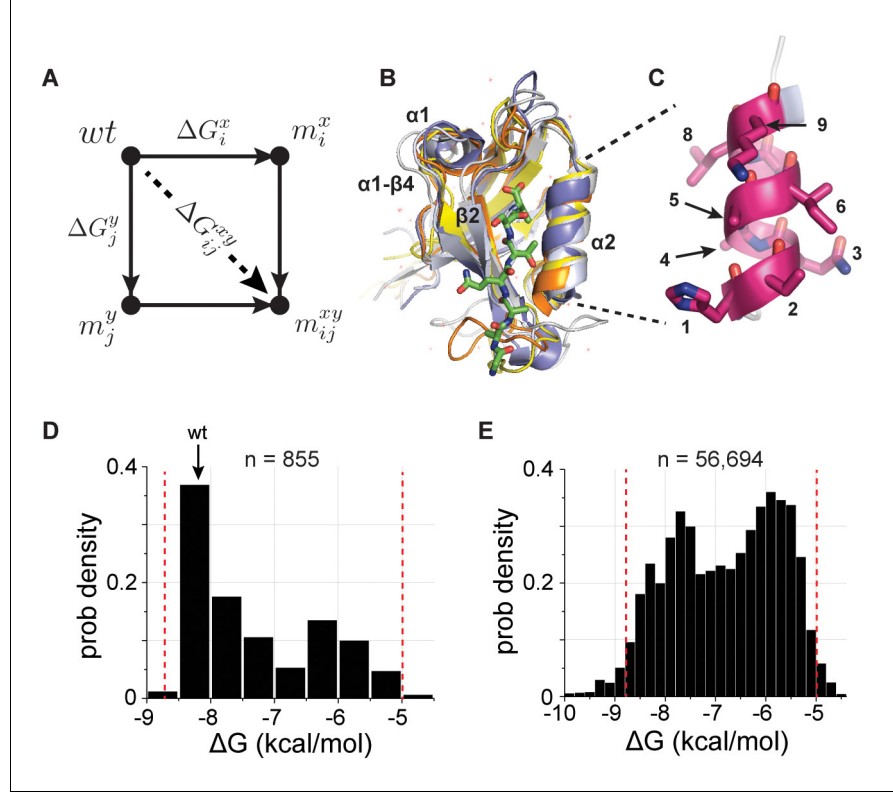

**Figure 1.** A deep coupling scan (DCS) for the PDZ binding pocket. (**A**), The thermodynamic double mutant cycle (TDMC), a formalism for studying the energetic coupling of pairs of mutations in a protein. Given two mutations ($x$ at position $i$ and $y$ at position $j$), the coupling free energy between them is defined as the extent to which the effect of the double mutation ($\Delta G_{ij}^{xy}$) is different from the summed effect of the mutations taken individually ($\Delta G_i^x + \Delta G_j^y$), a measure of the interaction (or epistasis) between the two mutations (see **Equation 1**, main text). (**B**), Structural overlay of the five PDZ homologs used in this study (PSD95[pdz3] (1BE9, white), PSD95[pdz2] (1QLC, orange), ZO1[pdz] (2RRM, yellow), Shank3[pdz] (5IZU, gray), and Syntrophin[pdz] (1Z86, blue)), emphasizing the conserved αβ-fold architecture of these sequence-diverse proteins (33% average identity, **Table 1**).Structural elements discussed in this work are indicated. (**C**), The nine-amino acid α2-helix, which forms one wall of the ligand-binding site. (**D–E**), The distribution of experimentally determined binding free energies, $\Delta G_{bind}$, for all single mutations (D, 855/855) and nearly all double mutations (E, 56,694/64,980) in the α2-helix for the 5 PDZ homologs, with the affinity of wild-type PSD95pdz3 indicated (wt). The red lines indicate the independently validated range of the assay (**Figure 1— figure supplement 1**); essentially all measurements fall within this range. These data comprise the basis for a deep analysis of conserved thermodynamic coupling in the PDZ family.

DOI: https://doi.org/10.7554/eLife.34300.002

The following figure supplements are available for figure 1:

**Figure supplement 1.** The bacterial two-hybrid assay for PDZ ligand binding.

DOI: https://doi.org/10.7554/eLife.34300.003

**Figure supplement 2.** Reproducibility and quality of the bacterial two-hybrid assay.

DOI: https://doi.org/10.7554/eLife.34300.004

$$\Delta\Delta G_{ij}^{xy} = \left( \Delta G_i^x + \Delta G_j^y \right) - \Delta G_{ij}^{xy}, \tag{1}$$

the difference between the effect predicted by the independent effects of the underlying single mutations and that of the actual double mutant. $\Delta\Delta G_{ij}^{xy}$ is typically proposed as an estimate for the degree of cooperativity between positions $i$ and $j$.

However, there are serious conceptual and technical issues with the usage of the TDMC formalism for deducing the energetic architecture of proteins. First, $\Delta\Delta G_{ij}^{xy}$ is not the coupling between the amino acids present in the wild-type protein (the 'native interaction'). It is instead the energetic

coupling *due to mutation*, a value that depends in complex and unknown ways on the specific choice of mutations made (*Faiman and Horovitz, 1996*). Second, global application of the TDMC method requires a scale of work matched to the combinatorial complexity of all potential interactions between amino acid positions under study. For even a small protein interaction module such as the PDZ domain (~100 residues, *Figure 1B*) (*Hung and Sheng, 2002*), a complete pairwise analysis comprising all possible amino acid substitutions at each position involves making and quantitatively measuring the equilibrium energetic effect of nearly two million mutations. Finally, even if these two technical issues were resolved, it is unclear how to go beyond the idiosyncrasies of one particular model system to the general, system-independent constraints that underlie protein structure, function, and evolvability.

Recent technical advances in massive-scale mutagenesis of proteins open up new strategies to address all these issues. In the PDZ domain, a bacterial two-hybrid (BTH) assay for ligand-binding coupled to next-generation sequencing enables high-throughput, robust, quantitative measurement of many thousands of mutations in a single experiment – a 'deep mutational scan' (*Fowler and Fields, 2014*; *McLaughlin et al., 2012*; *Raman et al., 2016*). Parameters of the BTH assay are tuned such that the binding free energy between each PDZ variant $x$ and cognate ligand ($\Delta G^x_{bind}$) is quantitatively reported by its enrichment relative to wild-type before and after selection ($\Delta E^x$, *Figure 1— figure supplement 1* and Materials and methods). This relationship enables extension of single mutational scanning to very large-scale double mutant cycle analyses – a 'deep coupling scan' (DCS) study (*Olson et al., 2014*). Indeed, the throughput of DCS is so high that it enables the study of double mutant cycles in several homologs of a protein family in a single experiment. Thus, DCS provides a first opportunity to deeply map the pattern and evolutionary conservation of interactions between amino acid residues in proteins, a strategy to reveal the fundamental constraints contributing to protein function.

Here, we apply DCS to several homologs of the PDZ domain family. The data show how to estimate native couplings from mutagenesis, and demonstrate the existence of an evolutionarily conserved network of cooperative amino acid interactions associated with ligand binding. We then use these data as a benchmark to test the predictive power of sequence-based coevolution methods, which if verified, would represent a general and scalable approach for defining the amino acid constraints underlying protein structure and function. We show that with different formulations, coevolution can indeed provide effective estimates of both structural contacts and cooperative functional interactions between residues. This work establishes a path towards a unified practical approach for understanding the design of natural proteins.

## Results

### A deep coupling scan in the PDZ family

To develop basic principles for high-throughput analysis of amino acid couplings, we focused on a region of the binding pocket of the PDZ domain, a protein-interaction module that has served as a powerful model system for studying protein energetics (*Lockless and Ranganathan, 1999*; *McLaughlin et al., 2012*). PDZ domains are mixed αβ folds that typically recognize C-terminal peptide ligands in a binding groove formed between the α2 and β2 structural elements (*Figure 1B*). We created a library of all possible single and double mutations in the nine-residue α2 helix of five sequence-diverged PDZ homologs (PSD95^pdz3, PSD95^pdz2, Shank3^PDZ, Syntrophin^PDZ, and Zo-1^PDZ, *Figure 1C*) (36 position pairs × 5 homologs, with 171 single + 12,996 double mutations + wild-type per homolog = 65,840 total variants) and measured the effect of every variant on binding its cognate ligand (*Figure 1D–E* and *Table 1*). Independent trials of this experiment show excellent reproducibility (*Figure 1—figure supplement 2*), and propagation of errors suggests an average experimental error in determining binding free energies of ~0.3 kcal/mol. Filtering for sequencing quality and counting statistics, we were able to practically collect 56,694 double mutant cycles (87% of total) for the α2 helix for all five homologs, with an average of 315 cycles per position pair per homolog (*Table 1*). Thus, we can (1) analyze the distributions of double mutant cycle coupling energies for nearly all pairs of mutations in the α2 helix and (2) study the divergence and conservation of these couplings over the five homologs.

**Table 1.** Summary of data collection.

For each PDZ homolog, we indicate the target ligand, the wild-type affinity, the top-hit sequence identity within the ensemble of homologs, and the assay/sequencing statistics.

| PDZ homolog | Ligand | Ligand sequence | Affinity | Top ID% (PDZ) | No. of single mutants (out of 171) | No. of double mutants (out of 12,996) | Mean cycles/position pair (out of 361) | Sequence readsunsel/sel |
|---|---|---|---|---|---|---|---|---|
| PSD95$^{pdz3}$ | CRIPT | TKNYKQTSV | 0.8 µM (*McLaughlin et al., 2012*) | 41.0 (Syntrophin$^{pdz}$) | 171 | 11,531 | 320 | 12,190,079/ 14,358,962 |
| PSD95$^{pdz2}$ | NMDAR2A | KMPSIESDV | 3.6 µM (*Stiffler et al., 2006*) | 40.5 (PSD95$^{pdz3}$) | 171 | 12,072 | 335 | 9,402,209/ 14,965,473 |
| Shank3$^{pdz}$ | Dlgap1/2/3 | YIPEAQTRL | 0.2 µM (*Stiffler et al., 2006*) | 25.0 (PSD95$^{pdz2}$) | 171 | 10,454 | 290 | 17,232,329/ 6,429,999 |
| Syntrophin$^{pdz}$ | Scn5a (Nav1.5) | PDRDRESIV | 1.6 µM (*Stiffler et al., 2006*) | 41.0 (PSD95$^{pdz3}$) | 171 | 10,757 | 298 | 8,227,200/ 15,248,680 |
| ZO-1$^{pdz}$ | Claudin8 | SIYSKSQYV | 4.6 µM (*Zhang et al., 2006*) | 37.5 (PSD95$^{pdz3}$) | 171 | 11880 | 330 | 5,365,041/ 11,523,044 |

DOI: https://doi.org/10.7554/eLife.34300.005

We first addressed the problem of how to estimate native coupling energies from mutant cycle data. In general, the effect of a mutation at any site in a protein is a complex perturbation of the elementary forces acting between atoms, with a net effect that depends on the residue eliminated, the residue introduced, and on any associated propagated structural effects. Thus, the distribution of thermodynamic couplings at any pair of positions over many mutation pairs could in principle be arbitrary and difficult to interpret. However, we find surprising simplicity in the histograms of coupling energies. In general, the data follow single or double-Gaussian distributions (*Figures 2* and *3*, *Figure 2—figure supplements 1–5*, and see Materials and methods), with most distributions centered close to zero and with just a few position pairs displaying two distinct populations. In general, every mutation is associated with the full range of coupling energies, and the distributions of couplings are not immediately obvious from known chemical properties of amino acids or secondary structure propensities (*Figure 3—figure supplement 1*). For example, mutations to glycine and proline might be expected to disrupt the α2 helix, and cause global large couplings with every other mutation, but in fact we find that these substitutions show a broad range of coupling energies not unlike other mutations. The data suggest that as an ensemble, mutations act as random perturbations to the native state of proteins, with the population-weighted mean of the distribution of coupling energies for each position pair (*Figures 2–3*, dashed lines) providing the best empirical estimate of the native interaction between amino acids through mutagenesis.

Two technical points are worth noting. First, the spread of the distributions is large, generally exceeding the estimated magnitude of the native interactions (*Figures 2–3*). This means (1) that traditional mutant cycle studies carried out with specific choices of mutations are more likely to just reflect the choice of mutations rather than the native interaction, and (2) that the only way to obtain good estimates of the native interaction between residues is to average over the effect of many double mutant cycles per position pair. The lack of such averaging could lead to considerable variation in the interpretation of mutant cycle data (*Chi et al., 2008*; *Faiman and Horovitz, 1996*; *Lockless and Ranganathan, 1999*). Second, we find that the BTH/sequencing approach displays such good reproducibility that it is possible to detect coupling energies with an accuracy that is on par with the best biochemical assays. For example, the average standard deviation in mean coupling energies for position pairs over four independent experimental replicates in PSD95$^{pdz3}$ is ~0.06 kcal/mol. Thus, we can map native amino acid interactions with high-throughput without sacrificing quality.

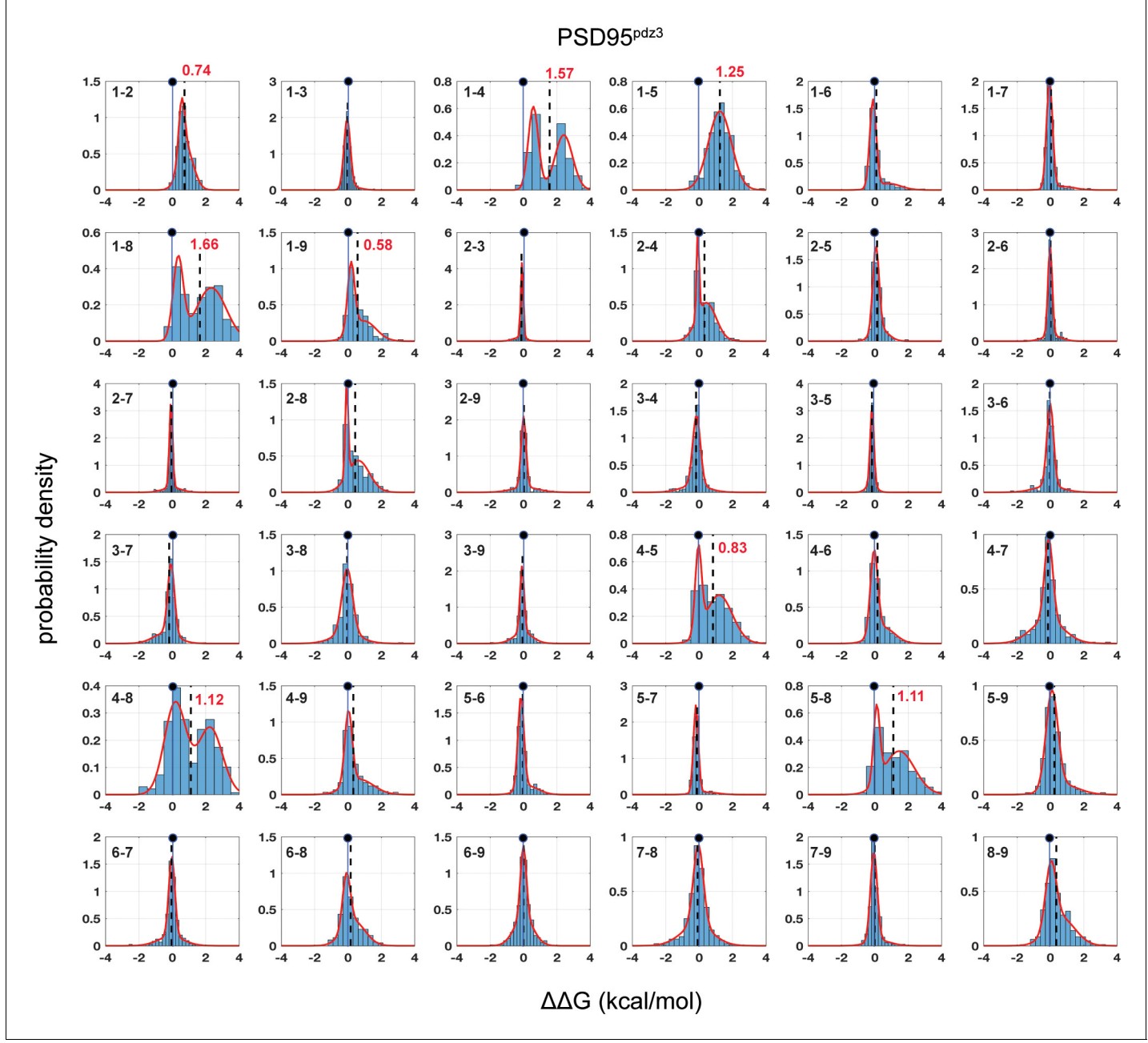

**Figure 2.** Distributions of pairwise thermodynamic couplings in a single PDZ homolog (PSD95[pdz3]). Each subplot shows the distribution of coupling free energies ($\Delta\Delta G$, see *Equation 1*, main text) for all measured mutants at one pair of positions in the α2-helix (numbering per *Figure 1C*) in PSD95[pdz3]. The distributions are fit to single or double Gaussians, using the Bayes Information Criterion to justify choice of model, and the position of zero coupling is indicated by the solid line and circle above. Population-weighted mean values are represented by dashed lines. The data are remarkably well defined by the fitted models. Most position pairs have distributions centered close to zero, with only eight pairs comprising all pairwise couplings between positions 1, 4, 5, and 8, and 1-2, 1-9 showing deviations. For these pairs, distributions of mutational coupling follow either a single mode (1-2, 1-5) or two modes with one centered at zero (1-4, 1-8, 1-9, 4-5, 4-8, 5-8); population-weighted mean values for these pairs are indicated in red.
*Figure 2—figure supplement 1–4* show similar data for each of the other homologs taken individually.
DOI: https://doi.org/10.7554/eLife.34300.006

The following figure supplements are available for figure 2:

**Figure supplement 1.** Distributions of double mutant cycles for each α2 helix position pair (numbering as in *Figure 1C*) for PSD95[pdz2].
DOI: https://doi.org/10.7554/eLife.34300.007

**Figure supplement 2.** Distributions of double mutant cycles for each α2 helix position pair (numbering as in *Figure 1C*) for Shank3[pdz].
DOI: https://doi.org/10.7554/eLife.34300.008

*Figure 2 continued on next page*

*Figure 2 continued*

**Figure supplement 3.** Distributions of double mutant cycles for each α2 helix position pair (numbering as in *Figure 1C*) for Syntrophin[pdz].
DOI: https://doi.org/10.7554/eLife.34300.009
**Figure supplement 4.** Distributions of double mutant cycles for each α2 helix position pair (numbering as in *Figure 1C*) for ZO1[pdz].
DOI: https://doi.org/10.7554/eLife.34300.010
**Figure supplement 5.** Distributions of coupling in relative enrichment ($\Delta\Delta E$) for a sampling of position pairs in the GB1 domain of the immunoglobulin-binding protein G.
DOI: https://doi.org/10.7554/eLife.34300.011

## A model for distributions of thermodynamic mutant cycle couplings

The uni/bi-modal character of distributions of thermodynamic mutant cycle couplings is striking in two respects. First is the generality. The same distribution shapes are found in all the individual PDZ homologs tested (*Figure 2* and *Figure 2—figure supplements 1–4*), the average over homologs (*Figure 3*), and even for DCS in an unrelated protein (GB1, *Figure 2—figure supplement 5* [*Olson et al., 2014*]). Second, the distribution shapes seem to be defined more by position, rather than by the character of mutations. For example, with a few exceptions, the same position-pairs in every PDZ homolog display mean coupling energies close to zero and the same few position pairs display bimodal or non-zero means (compare *Figure 2* and *Figure 2—figure supplements 1–4*, and see *Figure 3*). The sparse, position-specific character of bimodal distributions is also in the GB1 protein (*Figure 2—figure supplement 5*). These results imply a mechanism for the distributions of thermodynamic couplings in proteins that goes beyond local biophysical characteristics of the PDZ α2 helix or the average chemical properties of amino acids.

A simple mechanistic model for mutant cycle distributions is that the observed free energy of ligand binding arises from a cooperative internal equilibrium between two distinct conformational states of a protein (labeled 0 and 1, *Figure 4A*), with just a few sites defining this equilibrium. The basic idea is that any chemical reaction $K_x$ (here, binding) that is coupled to such an internal configurational equilibrium $K_c$ by a constant α will show an apparent equilibrium constant $K_x^{app}$ that is a distinct function of each of these three parameters. Specifically, $K_x^{app}$ depends linearly on $K_x$ (*Figure 4B*), displays a saturating relationship with non-trivial values of α (that is, for α >> 1) (*Figure 4D*), and depending on the degree of internal cooperativity, can show a sigmoidal or even ultrasensitive response to changes in $K_c$ (*Figure 4C*). The key to the bimodality lies in the nonlinearity of the relation between $K_x^{app}$ and the internal equilibrium $K_c$. With the wild-type value of $K_c$ set near to the non-linear region (that is, $K_c \sim 1$) and even without any intrinsic coupling in $K_x$ and α, it is straightforward to see that mutations perturbing only $K_x$ and α will generate distributions of thermodynamic couplings centered at zero (*Figure 4E*), but perturbations in $K_c$ can evoke bimodal distributions with one mode centered at zero (*Figure 4F*) or a single distribution centered at a non-zero value (*Figure 4G*). With slight variations in the wild-type value of $K_c$ between homologs, this model can account for all the observed distributions of pairwise thermodynamic coupling reported here. In addition, the sparse, position-specific character of bimodal or non-zero couplings arises from the constraint that only a few cooperative positions in the protein control the internal conformational equilibrium ($K_c$).

We note that the model is intended at this stage as a hypothesis rather than proof of mechanism. Nevertheless, we note that a cooperative two-state internal equilibrium involving the α2 helix has been experimentally observed in a PDZ domain, and is part of an allosteric regulatory mechanism controlling ligand binding (*Mishra et al., 2007*). Specifically, in the *Drosophila* InaD protein, redox-dependent regulation of $K_c$ in one PDZ domain switches the conformation of the ligand binding pocket and controls the dynamics of visual signaling (*Helms, 2011*; *Mishra et al., 2007*). The findings here of bimodality in mutational couplings in diverse PDZ homologs and in the GB1 protein suggests that a two-state internal equilibrium may be a common feature of many proteins. If so, the residues defining $K_c$ may represent the mechanistic basis for classic thermodynamic concept of allosteric regulation in proteins through modulation of two-state conformational equilibria (*Cui and Karplus, 2008*; *Monod et al., 1965*; *Volkman et al., 2001*).

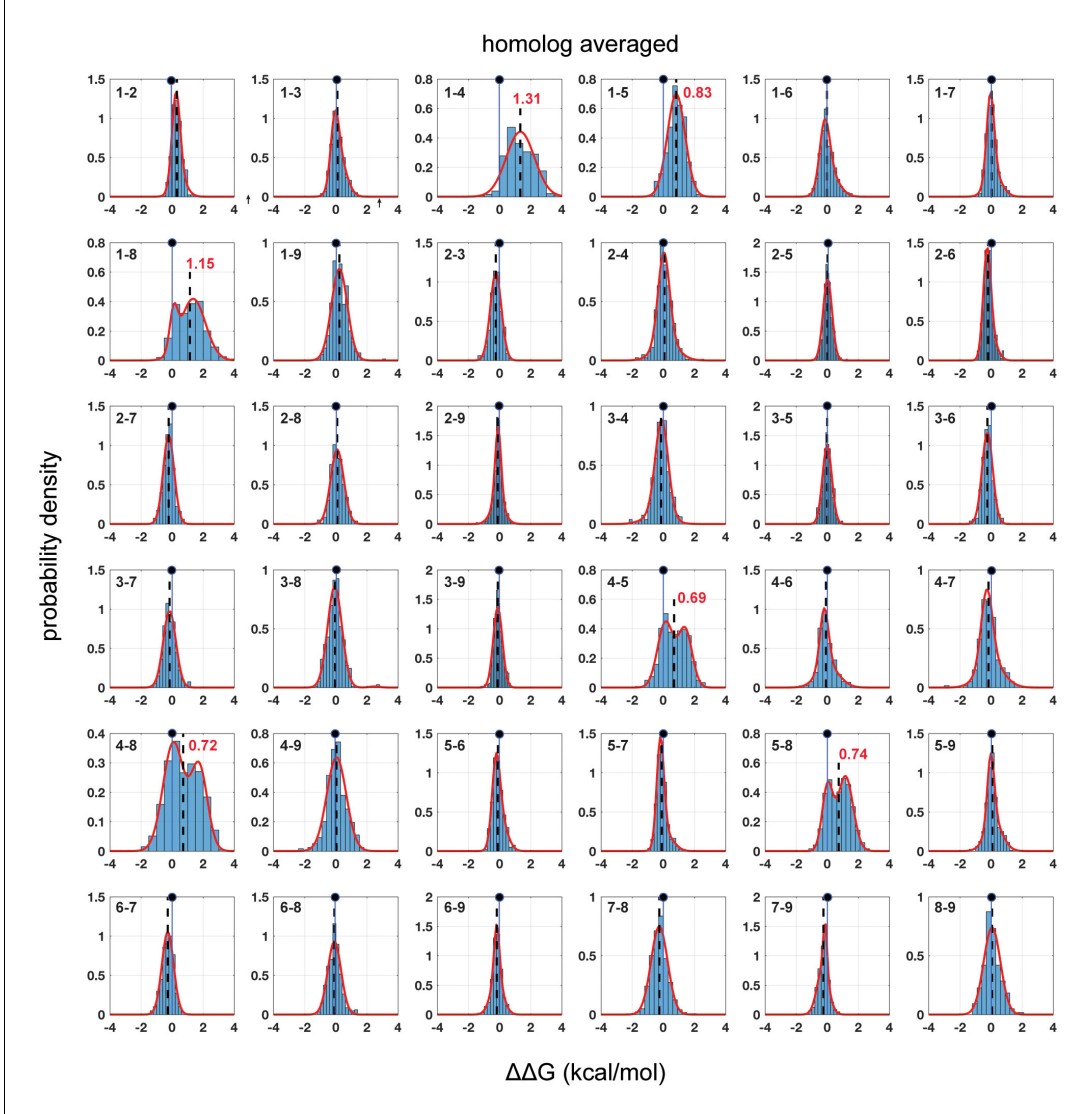

**Figure 3.** Homolog-averaged pairwise thermodynamic couplings in the PDZ domain. Each subplot shows the distribution of coupling free energies ($\Delta\Delta G$, see *Equation 1*, main text) for all measured mutants at one pair of positions in the α2-helix, but here averaged over the five homologs. As in *Figure 2*, the distributions are fit to single or double Gaussians, using the Bayes Information Criterion to justify choice of model. The position of zero coupling is indicated by the solid line and circle above and population-weighted mean values are represented by dashed lines. Averaging over homologs reveals the conserved pattern of couplings; now, only six pairs comprising all pairwise couplings between positions 1, 4, 5, and 8 show deviations from zero. For these pairs, distributions of mutational coupling follow either a single mode (1-4, 1-5) or two modes with one centered at zero (1-8, 4-5, 4-8, 5-8); population-weighted mean values for these pairs are indicated in red.

DOI: https://doi.org/10.7554/eLife.34300.012

The following figure supplement is available for figure 3:

**Figure supplement 1.** Amino acid contributions to distributions of double mutant cycle coupling energies for the PDZ α2 helix.
DOI: https://doi.org/10.7554/eLife.34300.013

## Idiosyncrasy and conservation in functional couplings in the PDZ domain

What do the data tell us about the overall pattern of amino acid interactions? *Figure 5A–E* show heat maps of the estimated native coupling energies between all pairs of amino acids within the α2 helix for each PDZ homolog. The data demonstrate both idiosyncrasy and conservation of amino acid couplings in paralogs of a protein family. For example, helix positions 3–4 show moderate couplings in two of the domains (PSD95[pdz3] and Syntrophin[PDZ], *Figure 5A and D*) but not in the other

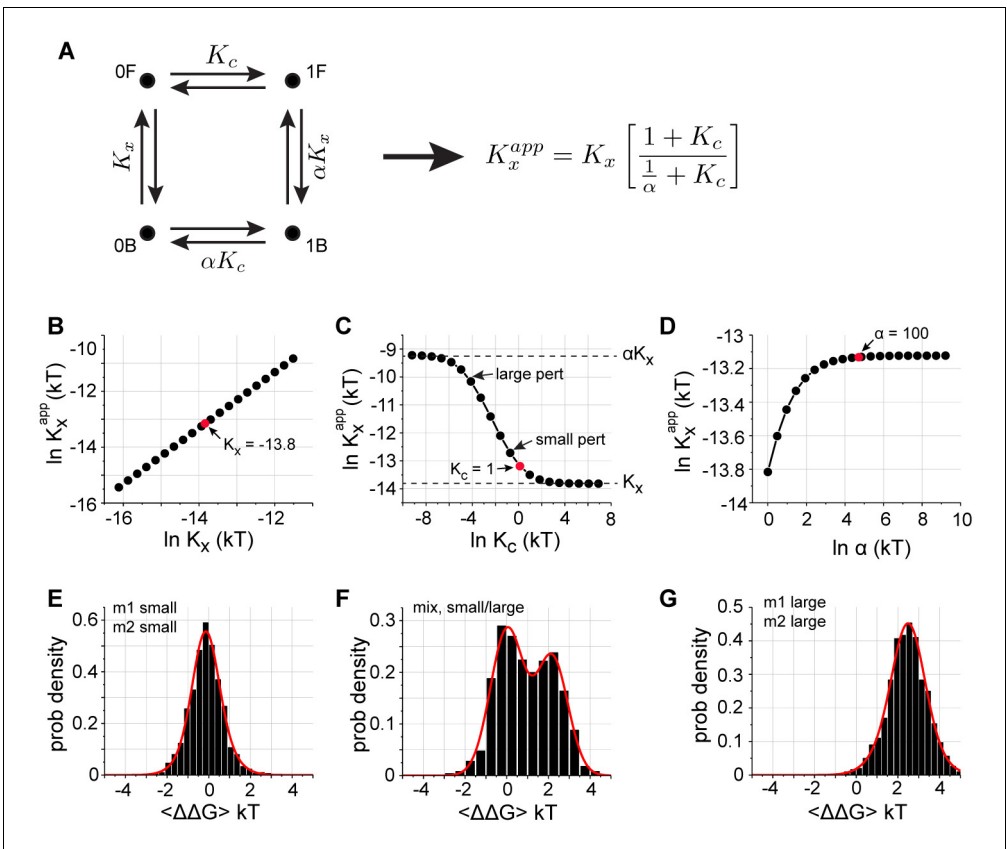

**Figure 4.** A basic model for observed distributions of double mutant cycle coupling energies. (**A**), A schematic representation of two coupled equilibria in a protein molecule – a reaction with equilibrium constant $K_x$ corresponding to function (here, binding), an internal two-state conformational equilibrium defined by $K_c$, and a coupling parameter $\alpha$ linking the two. The equation at right shows the general analytic solution for how the apparent equilibrium constant $K_x^{app}$ depends on these three parameters, and panels (**B–D**) show graphs of these relationships over a relevant range of values. Note that $K_x$ (and $\alpha K_x$) are defined as dissociation constants, and $K_c \equiv [0F]/[1F]$ and $\alpha K_c \equiv [0B]/[1B]$. (**B–D**), $K_x^{app}$ shows a linear dependence on $K_x$, a saturating relationship with $\alpha$, and a sigmoidal relationship with $K_c$. For a range of $K_c$, $K_x^{app}$ ranges between $K_x$ and $\alpha K_x$, the two extreme limits set by the reaction diagram in panel (**A**). (**E–G**), distributions of coupling energies for simulations in which we choose a set of 'wild-type' values of $K_x$, $K_c$, and $\alpha$ (red dots, panels **B–D**) and consider mutations that cause random Gaussian perturbations of $K_x$ and $\alpha$, but either small or large perturbations of $K_c$ (indicated in panel **C**). If all mutations cause small effects in $K_c$, we obtain unimodal distributions centered at zero coupling energy (**E**), and if all mutations cause large effects in $K_c$, we obtain unimodal distributions centered at a non-zero coupling energy (**G**). However, if mutations cause a mix of small and large effects on $K_c$, we obtain bimodal distributions with one mode centered at zero (**F**). These three types recapitulate all the observed distributions for all PDZ homologs (main *Figure 2* and *Figure 1—figure supplement 1–4*), for the GB1 protein (*Figure 1—figure supplement 5*), and for the average over homologs (*Figure 3*). Note that higher order cooperativity between amino acids specifying $K_c$ (a plausible scenario), would further steepen the relationship shown in panel (**C**) and would cause the all-or-nothing character of mutations with regard to $K_c$ with even less distinction between large and small perturbations. This model is not intended as a proof of mechanism for the observed distributions, but instead provides a logical scheme that explains the observations in light of known two-state allosteric equilibria is some PDZ domains (*Mishra et al., 2007*; *Raman et al., 2016*).

DOI: https://doi.org/10.7554/eLife.34300.014

homologs. Similarly, coupling between positions 7–8 is shared by PSD95[pdz3], PSD95[pdz2], and Zo1[PDZ] (*Figures 5A, B and E*) but not in the other two homologs. In contrast, all pairwise interactions between positions 1, 4, 5, and 8 show a systematic pattern of energetic coupling in all homologs tested. Thus, each PDZ domain displays variations in the pattern and strength of amino acid energetic couplings, but also includes a set of evolutionarily conserved couplings at a few positions. We

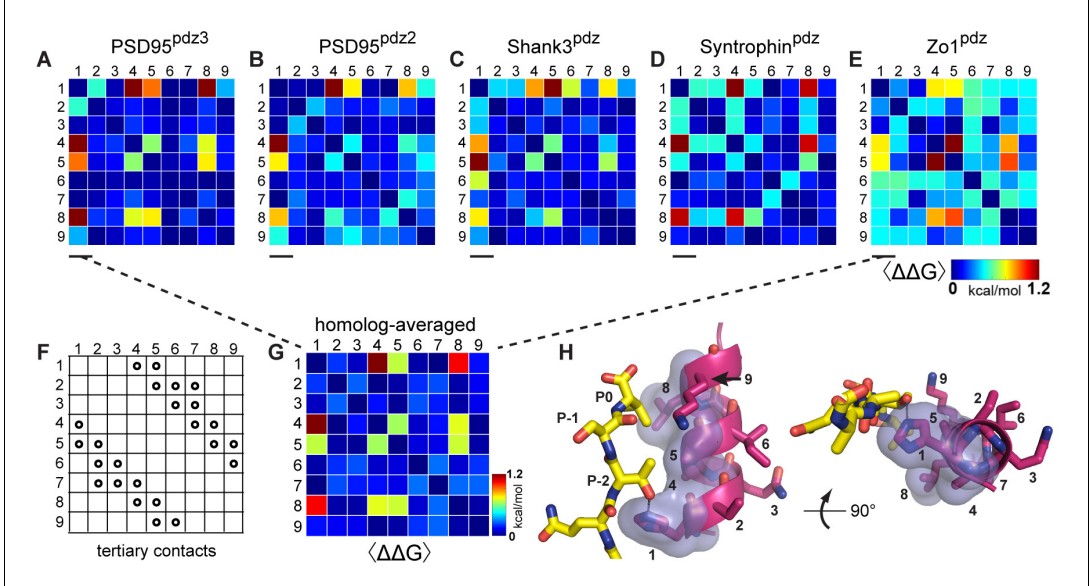

**Figure 5.** Conservation and idiosyncrasy in the pattern of energetic couplings over PDZ homologs. (A–E), Matrices of mutation averaged pairwise thermodynamic couplings for the α2-helix in each PDZ homolog. The color scale is chosen to represent the full range of measured energetic couplings. The data show that some couplings are specific to individual homologs or shared by a subset of homologs, but that couplings between positions 1, 4, 5, and 8 are conserved over homologs. (F), the pattern of direct tertiary contacts between amino acid positions in the PDZ α2 helix. By convention (*Morcos et al., 2011*), trivial contacts between residues with sequence distance less than three are not shown. (G), The homolog and mutation averaged couplings (corresponding to *Figure 3*), displaying the conserved interactions between amino acids in the PDZ α2-helix. (H), Two views of the α2-helix, with the four interacting positions in the homolog-averaged dataset shown in transparent surface representation, and ligand in yellow stick bonds. These include three positions in direct contact with ligand (1, 5, 8) and one allosteric position buried in the core of the protein (4).
DOI: https://doi.org/10.7554/eLife.34300.015

take the conserved couplings to represent the most fundamental constraints underlying PDZ function, with the homolog-specific couplings indicating more specialized or even serendipitous couplings.

To isolate the fundamental couplings, we averaged all the double mutant cycle data over all mutations and over the five PDZ homologs tested (*Figure 3*), resulting in a matrix of evolutionarily conserved pairwise thermodynamic couplings (*Figure 5G*). This analysis reinforces the result that positions 1, 4, 5, and 8 comprise a cooperative network of functional residues in the PDZ domain family, and the remainder, even if in direct contact with each other or with ligand, contribute less and interact idiosyncratically or not at all. The conserved couplings form a chain of physically contiguous residues in the tertiary structure that both contact (1, 5, 8) and do not contact (4) the ligand (*Figure 5H*). Interestingly, position 4 is part of the distributed allosteric regulatory mechanism in the InaD PDZ domain discussed above (*Mishra et al., 2007*), providing a biological role for its energetic connectivity with binding pocket residues. Overall, the pattern of couplings does not just recapitulate all tertiary contacts between residues (compare *Figure 5F* with *Figure 5G*) or the pattern of internal backbone hydrogen bonds that define this secondary structure element. Instead, conserved amino acid interactions in the PDZ α2 helix are organized into a spatially inhomogeneous, cooperative network that underlies ligand binding and allosteric coupling.

The salient point that emerges from these data is that the pattern of direct contacts that define the protein structure and the pattern of cooperative amino acid interactions that define protein function are not the same. Both coexist and are relevant, but represent distinct aspects of the energetic architecture of proteins.

## Coevolution-based inference of functional couplings

This result begins to expose the complex energetic couplings underlying protein function, but also highlights the massive scale of experiments required to deduce this information for even a few amino acid positions. How then can we practically generalize this analysis to deduce all amino acid

interactions in a protein, and for many different proteins? There are potential strategies for pushing deep mutational coupling to larger scale, but quantitative assays such as the BTH are difficult to develop, mutation libraries grow exponentially with protein size, and the averaging over homologs will always be laborious, expensive, and incomplete. In addition, the cooperative action of amino acids could contribute both positive (*McLaughlin et al., 2012*) and negative design (*Noivirt-Brik et al., 2009*) features in proteins, and it is often not easy to create high-throughput assays for measuring all aspects of proteins that make up function.

A different approach is suggested by understanding the rules learned in this experimental study for discovering relevant energetic interactions within proteins. The bottom line is the need to apply two kinds of averaging. Averaging over many mutations provides an estimate of native interaction energies between positions, and averaging the mutational effects over an ensemble of homologs separates the idiosyncrasies of individual proteins from that which is conserved in the protein family. Interestingly, these same rules also comprise the philosophical basis for a class of methods for estimating amino acid couplings through statistical analysis of protein sequences. The central premise is that the relevant energetic coupling of two residues in a protein should be reflected in the correlated evolution (coevolution) of those positions in sequences comprising a protein family (*Göbel et al., 1994*; *Lockless and Ranganathan, 1999*; *Neher, 1994*; *Weigt et al., 2009*). Statistical coevolution also represents a kind of combined averaging over mutations and homologs, and if experimentally verified, would (unlike deep mutational studies) represent a scalable and general approach for learning the architecture of amino acid interactions underlying function in a protein. The data collected here provides the first benchmark data to deeply test the predictive power of coevolution-based methods.

One approach for coevolution is the statistical coupling analysis (SCA), a method based on measuring the conservation-weighted correlation of positions in a multiple sequence alignment, with the idea that these represent the relevant couplings (*Halabi et al., 2009*; *Lockless and Ranganathan, 1999*). In the PDZ domain family (~1600 sequences, pySCA6.0 [*Rivoire et al., 2016*]), SCA reveals a sparse internal organization in which most positions evolve in a nearly independent manner and a few (~20%) are engaged in a pattern of mutual coevolution (*Halabi et al., 2009*; *Lockless and Ranganathan, 1999*; *Rivoire et al., 2016*). In this case, the coevolving positions are simply defined by the top eigenmode (or principal component) of the SCA coevolution matrix. Extracting the corresponding coevolution pattern for just the α2 helix (*Figure 6*), we find that coevolution as defined by SCA strongly predicts the homolog-averaged experimental couplings collected here in a manner robust to both alignment size and method of construction ($r^2 = 0.82 - 0.77$, $p = 10^{-14} - 10^{-12}$ by F-test, indicating the significance of the correlation coefficient, *Figures 6B–D* and *Figure 6—figure supplement 1*). The predictions also hold for individual homologs (*Figure 6—figure supplement 2*), consistent with the premise that the essential physical constraints underlying function are deeply conserved. Importantly, the goodness of prediction depends strongly on both of the basic tenets that underlie the SCA method – conservation-weighting (*Figure 6E–F*) and correlation (*Figure 6G–H*) (*Rivoire et al., 2016*).

A basic result of the SCA method is that groups of coevolving positions form physically connected networks of amino acids (termed protein 'sectors') that link the main functional site to distantly positioned allosteric sites (*Halabi et al., 2009*; *Lockless and Ranganathan, 1999*; *Süel et al., 2003*). Indeed, in the PDZ domain, the protein sector represents a chain of amino acids the links the β2-β3 loop with the α1-β4 surface through the binding pocket and the buried α1 helix (spheres and surface, *Figure 7*). The α1-β4 surface is a known binding site for allosteric modifiers (*Peterson et al., 2004*), and the β2-β3 loop contains positions where mutations can enable adaptation to new ligand specificities (*Raman et al., 2016*). The four positions experimentally identified here as a cooperative unit (1, 4, 5, 8, red spheres, *Figure 7*) represent the portion of the α2 helix that is contained in the protein sector. Thus, these data argue that the sector correctly identifies the amino acids engaged in cooperative interactions, but more importantly implies that these positions are just a part of a more global cooperative unit within the PDZ domain that mediates allosteric communication.

Another approach for amino acid coevolution is direct contact analysis (DCA, [*Marks et al., 2011*; *Morcos et al., 2011*]), a method developed for the prediction of tertiary contacts in protein structures. DCA uses classical methods in statistical physics to deduce a matrix of minimal pairwise couplings between positions ($J_{ij}$, *Figure 8A*) that can account for the observed correlations between amino acids in a protein alignment, with the hypothesis that the strong couplings in $J_{ij}$ will be direct

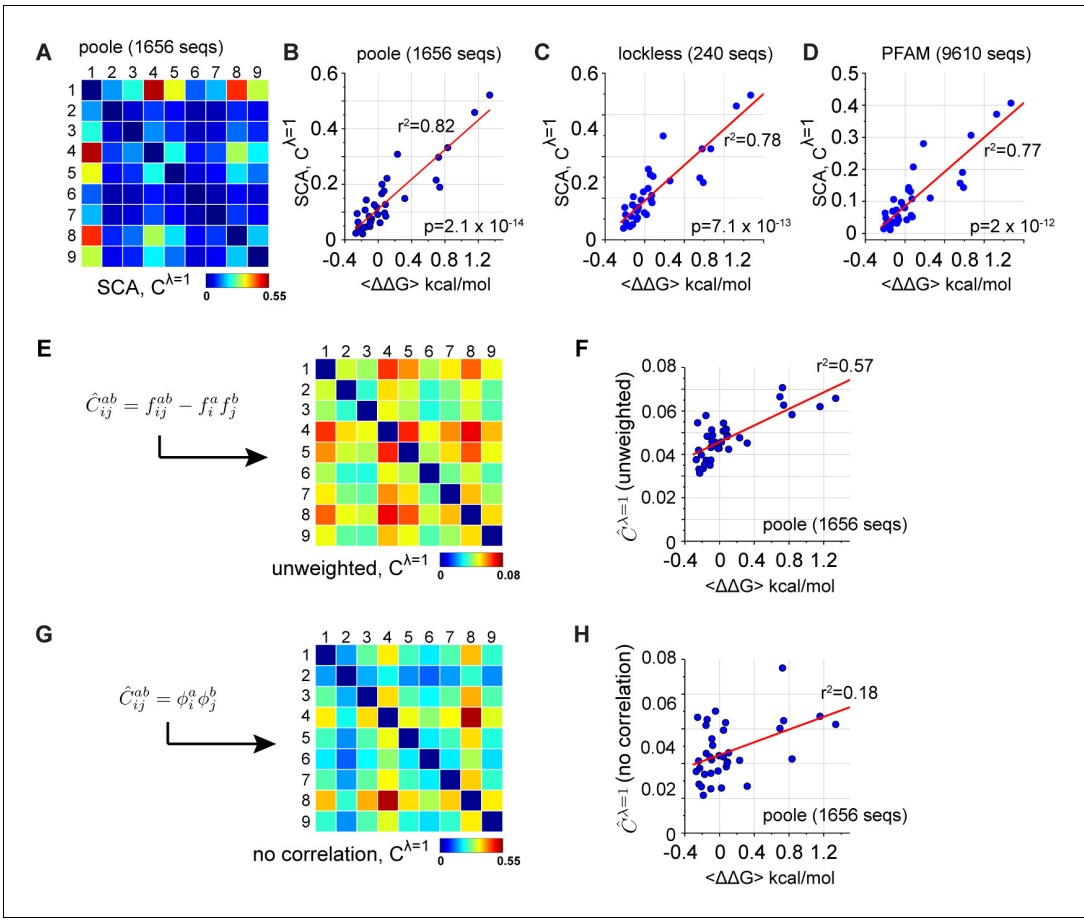

**Figure 6.** Coevolution-based inference of energetic couplings - SCA. (**A**), Coevolution of sequence positions corresponding to the top eigenmode of the SCA matrix, derived from an alignment of 1656 eukaryotic PDZ domains (the 'Poole' alignment). The data show that a subset of positions coevolve within the PDZ α2-helix. (**B–D**), The relationship between experimental homology-averaged energetic couplings ($\langle\Delta\Delta G\rangle$) and SCA-based coevolution computed for three different alignments that differ in size and method of construction. The p-values give the significance of the coefficient of determination ($r^2$) by the F-test. (**E–H**), The basic calculation in SCA is to compute a conservation-weighted correlation matrix $\tilde{C}_{ij}^{ab} = \phi_i^a \phi_j^b \left[ f_{ij}^{ab} - f_i^a f_j^b \right]$, where $f_i^a$ and $f_{ij}^{ab}$ represent the frequency and joint frequencies of amino acids $a$ and $b$ at positions $i$ and $j$, respectively, in a multiple sequence alignment. The term $f_{ij}^{ab} - f_i^a f_j^b$ gives the correlation of amino acids at each pair of positions, and $\phi$ represents a weighting function for each amino acid at each position that is related to its conservation (*Halabi et al., 2009*; *Rivoire et al., 2016*). We compared the relationship of the experimental energetic couplings ($\langle\Delta\Delta G\rangle$) with measures of coevolution that leave out the conservation weights (**E–F**), or that leave out the correlations (**G–H**). The analysis shows that both terms contribute to predicting native energetic couplings between amino acids.
DOI: https://doi.org/10.7554/eLife.34300.016

The following figure supplements are available for figure 6:

**Figure supplement 1.** Robustness of the SCA to alignment size.
DOI: https://doi.org/10.7554/eLife.34300.017

**Figure supplement 2.** The relationship between mutation-averaged couplings and predictions from SCA for individual domains.
DOI: https://doi.org/10.7554/eLife.34300.018

---

contacts in the tertiary structure. Indeed, studies convincingly demonstrate that the top $L/2$ (where L is the length of the protein) couplings are highly enriched in direct structural contacts (*Anishchenko et al., 2017*). Consistent with this, this method successfully identifies direct contacts in the PDZ α2 helix (*Figure 8A*, compare heat map to white and black circles) to an extent that agrees with the reported work. However, DCA model makes predictions of functional energetic couplings

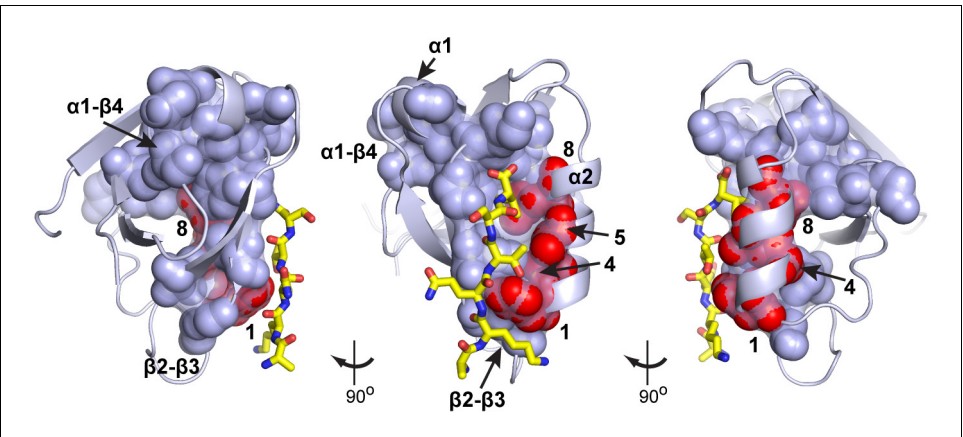

**Figure 7.** Homolog-averaged thermodynamic couplings and protein sectors. Analysis of the top eigenmodes of the SCA coevolution matrix exposes groups of coevolving amino acids that empirically are found to form physically contiguous networks in the tertiary structure, often connecting the main functional site to remote allosteric sites (*Halabi et al., 2009*; *Lockless and Ranganathan, 1999*; *Rivoire et al., 2016*; *Süel et al., 2003*). In the PDZ family, the protein sector (shown as CPK spheres and transparent surface on three rotations of a representative structure, PDB 1BE9) connects the ligand binding pocket to two known allosteric sites, one in the α1-β4 loop (*Peterson et al., 2004*) and the other in the β2-β3 loop (*Raman et al., 2016*); the ligand is shown in yellow stick bonds. The homolog-averaged thermodynamic couplings in the α2 helix (positions 1, 4, 5, and 8) precisely correspond to the portion of the PDZ sector contributed by the secondary structure element. The selective cooperative action of these residues is consistent with the idea that the sector represents a global collective mode in the PDZ structure associated with function, embedded within a more independent environment.

DOI: https://doi.org/10.7554/eLife.34300.019

between mutations (*Figure 8B*) that are weakly or not at all related to the homolog-averaged experimental data ($r^2 = 0.33 - 0.05$, $p = 10^{-3} - 0.09$ by F-test, *Figure 8C–E*). Interestingly, the best predictive power comes from one moderately-sized structure-based sequence alignment (*Figure 8C*) rather than from the largest publicly available alignments (*Figure 8D–E*). These results are similar or poorer for prediction of couplings in individual domains (*Figure 8—figure supplement 2*). Due to inclusion of many unconserved correlations, DCA is quite sensitive to alignment size, with random sub-samplings of the best performing alignment producing models with variable quality in terms of predicting the data (*Figure 8—figure supplement 1*).

The top couplings in the $J_{ij}$ matrix identify local structural contacts between amino acids, but do these direct couplings also underlie the partial ability of DCA to account for functional couplings? To test this, we chose the best-case alignment (the 'Poole' alignment, *Figure 8C*), and made an edited DCA model in which only the top L/2 pairwise couplings in $J_{ij}$ that define tertiary contacts are retained and the remaining weaker non-contacting couplings are randomly scrambled. While the full model shows moderate association with experimental data ($r^2 = 0.33$, $p = 10^{-3}$ by F-test, *Figure 8C*), the edited model shows predictions that are now unrelated to the experimental data ($r^2 = 0.02$, $p = 0.21$ by F-test, *Figure 8F*). Thus, the many non-contact pairwise couplings in the DCA model, which represent noise from the point of view of structure prediction, contribute significantly to prediction of function. A similar result has been noted in the DCA-based prediction of protein-protein interactions, where the quality of prediction depends on many weak couplings between residues not making contacts at the interface (A.F. Bitbol and N. Wingreen, personal communication).

A recent study has shown that for the strong couplings in the DCA model (the top L/2 terms in $J_{ij}$, *Figure 8B*), cases of apparently non-contacting residues are often resolved as true contacts by one of three explanations: (a) they are contacts induced by oligomerization, (b) they are contacts in other conformational states or homologous structures, or (c) they are artifacts due to misalignment of repeat regions (*Anishchenko et al., 2017*). With the presumption that all the weaker remaining terms in $J_{ij}$ are irrelevant, this result has been interpreted to mean that all the evolutionary constraint in protein structures is in direct physical contacts, with allosteric mechanisms not contributing to

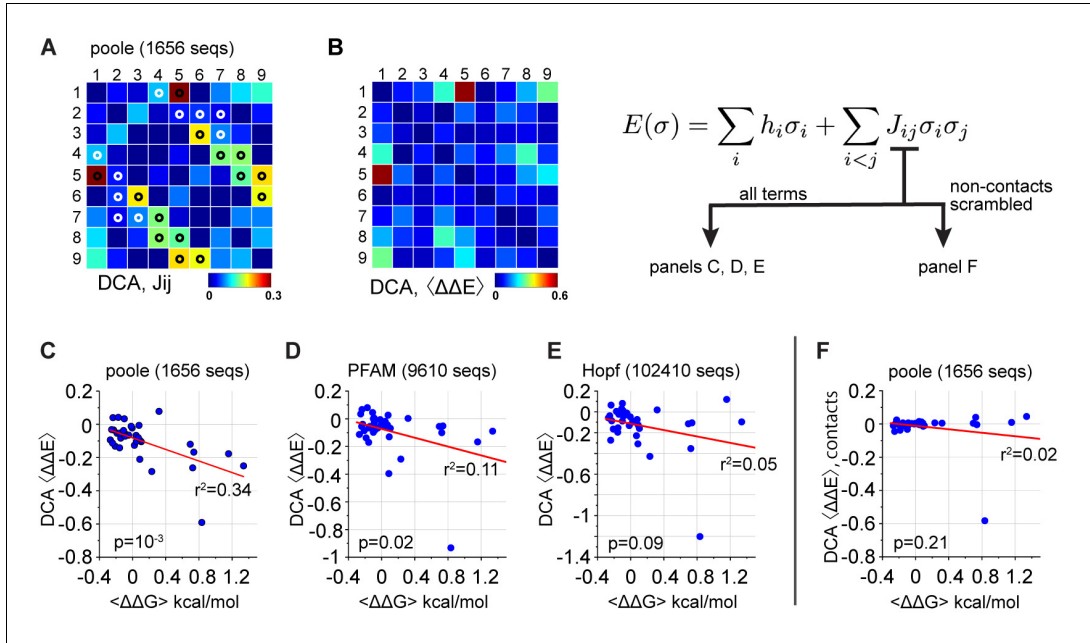

**Figure 8.** Coevolution-based inference of energetic couplings - DCA. (A), The matrix of direct couplings ($J_{ij}$) from the DCA method, with tertiary contacts in the PDZ structure (1BE9) indicated by white or black circles. By convention (*Morcos et al., 2011*), trivial contacts between residues with sequence distance less than three are not shown. The data show that all top direct couplings identified by DCA are indeed tertiary structural contacts. (B), The DCA method involves the inference of a statistical energy function $E(\sigma)$ that for each sequence $\sigma$, is parameterized by a set of intrinsic constraints on amino acids ($h_i$) and pairwise interactions between amino acids ($J_{ij}$). These parameters are optimized to reproduce the observed alignment frequencies and pairwise correlations. Using the model, the matrix shows mutation- and homolog-averaged energetic couplings, computed precisely as for the experimental data; see Materials and methods for details. (C–E), The relationship between experimental ($\langle\Delta\Delta G\rangle$) and DCA-inferred ($\langle\Delta\Delta E\rangle$) couplings in the PDZ α2-helix, for three PDZ alignments that differ in size and method of construction. The p-values give the significance of the coefficient of determination ($r^2$) by the F-test. (F), The relationship between experimental and DCA-inferred couplings from $J_{ij}$ in which top couplings defining contacts are preserved and all non-contact couplings are randomly scrambled. The DCA model used for this analysis is from the Poole alignment, as in panel D. The data show that pairwise couplings in the DCA model between non-contacting positions contribute significantly to prediction of protein function.
DOI: https://doi.org/10.7554/eLife.34300.020

The following figure supplements are available for figure 8:

**Figure supplement 1.** Robustness of DCA to alignment size.
DOI: https://doi.org/10.7554/eLife.34300.021

**Figure supplement 2.** The relationship between mutation-averaged couplings and predictions from DCA for individual domains.
DOI: https://doi.org/10.7554/eLife.34300.022

coevolution (*Anishchenko et al., 2017*). In one sense, the data shown here are fully consistent with the results of this previous study; the top terms in $J_{ij}$ are indeed enriched in contacts (*Figure 8A*), and in fact do not correspond to the experimental energetic couplings (*Figure 8F*), including long-range ones such as 1-8 (*Figure 5G*). But, the finding that the weak, non-contacting couplings in $J_{ij}$ contribute to predicting the experimental data argue that the origin of evolutionary constraints is not strictly in direct physical interactions of amino acids. Furthermore, the finding that the SCA coevolution matrix provides an improved prediction of experimental couplings (*Figure 8B–D*), including long-range ones (e.g. 1-8, *Figure 6A*) argues that information about allosteric energetic interactions are contained in the statistics of alignments and are therefore part of the total evolutionary constraint.

Taken together, these findings provide a set of important clues for now extending the statistical physics approach to produce models for proteins that accurately predict interactions that define both local structural contacts and the global, collective actions of residues that underlie function.

## Discussion

Defining the pattern of cooperative interactions between amino acids is essential for understanding the evolutionary design of protein structure and function. Here, we use very high-throughput next-generation sequencing based mutagenesis to experimentally probe the pattern of functional interactions between residues. We show that averaging thermodynamic couplings over many pairs of mutations provides an estimate of the native interactions between amino acids, and exposes an architecture in which most pairs of amino acids are uncoupled and a few significantly interact to make a cooperative network underlying function. Further averaging over homologs refines the pattern of cooperativity, revealing an evolutionarily conserved network of cooperative amino acid interactions that includes both direct and allosteric influences on ligand binding. This pattern is distinct from the pattern of local contacts between residues that defines secondary structure elements and the tertiary structure, indicating that a full understanding of proteins requires inference of both direct local structural contacts and the network of cooperative interactions that underlies function.

While the DCS method represents a productive extension of 'deep mutagenesis' methods to probe second-order cooperative interactions between amino acids, the combinatorial complexity of cooperative amino acid interactions is so vast that no experiment can exhaustively probe the global pattern of amino acid interactions within proteins. In this regard, we suggest that DCS serves mainly to provide a critical benchmark to explore other strategies that have the generality and scalability to learn the global pattern. Such a strategy is statistical coevolution, the concept that the relevant energetic interactions between amino acids contributing to structure and function should be reported in the correlations of amino acid outcomes at pairs of positions in a large sampling of homologous sequences comprising a protein family. In fact, we show that two different approaches for coevolution – DCA and SCA – effectively report the experimentally determined pattern of structural contacts and functional couplings, respectively. While prediction of structural contacts are easily verified by comparison with published protein structures (*Kamisetty et al., 2013*; *Marks et al., 2012*), datasets for evaluating the prediction of protein function have been limited (*Lockless and Ranganathan, 1999*; *McLaughlin et al., 2012*; *Teşileanu et al., 2015*). In this regard, DCS represents a necessary step for collecting the kind of data to refine and test models for protein function.

The finding that SCA can effectively predict the conserved thermodynamic couplings allows us to propose a deeper hypothesis about the meaning and role of protein sectors (*Halabi et al., 2009*). The coupled equilibrium model described above (*Figure 4A*) postulates the existence of a cooperative two-state internal equilibrium $K_c$ within proteins, where only perturbations of $K_c$ can generate non-zero mutational couplings. Since significant conserved couplings in the α2 helix are exclusively within positions 1, 4, 5, and 8 and since these positions are contained within the protein sector, it is logical to propose that the sector represents the structural unit underlying $K_c$ – a distributed cooperative amino acid network through which allosteric effects can be transmitted by modulation of the internal conformational equilibrium. Consistent with this, introduction of new molecular interactions at sector edges has been shown to be a route to engineering new allosteric control in protein molecules (*Lee et al., 2008*; *Reynolds et al., 2011*). In future work, it will be interesting to rigorously test the hypothesis that the sector underlies $K_c$ through global or sector-directed DCS experiments.

Overall, our findings clarify the current state of sequence-based inference of protein structure and function (*Figliuzzi et al., 2016*; *Hopf et al., 2017*). DCA successfully predicts contacts in protein structures in the top couplings, but in its current form, does not appear to capture the cooperative constraints that underlie protein function well. In contrast, SCA does not predict direct structural contacts well, but instead seems to accurately capture the energetic couplings that contribute to protein function. As explained previously, these two approaches sample different parts of the information contained in a sequence alignment (*Cocco et al., 2013*; *Rivoire, 2013*), and therefore are not mutually incompatible. These results highlight the need to unify the mathematical principles of contact prediction and SCA-based energetic predictions towards a more complete model of information content in protein sequences.

In summary, the collection of functional data for some 56,000 mutations in a sampling of PDZ homologs demonstrates an evolutionarily conserved pattern of amino acid cooperativity underlying function. This pattern is well-estimated by statistical coevolution based methods, suggesting a powerful and (given the scale of experiments necessary) uniquely practical approach for mapping the architecture of couplings between amino acids. Indeed, the remarkable implication is that with a sufficient ensemble of sequences comprising the evolutionary history of a protein family and the further technical advancements suggested above, the pattern of relevant amino acid interactions can be inferred without any experiments.

## Materials and methods

### Library generation

For each PDZ homolog, a library of all single and pairwise mutations in the α2-helix was generated using a set of 36 mutagenic forward primers (50-60mers, IDT), each with two codons randomized as NNS (IUPAC code). Each primer was used in a separate inverse PCR reaction with a constant reverse primer, a PZS22 plasmid containing the wildtype PDZ variant as template, and Q5 polymerase (NEB). The primers amplify the entire plasmid, introducing mutations only on one strand, a strategy that reduces library bias and over-representation of the wildtype allele that occurs with methods such as overlap-extension PCR (*Jain and Varadarajan, 2014*). Both forward and reverse primers are designed with BsaI sites in the 5' region, permitting scarless unimolecular ligation of the PCR products. All 36 PCR products per PDZ homolog are quantified by Qubit and Nanodrop (Thermo Fisher Scientific), mixed in an equimolar ratio, and used for a one-pot digestion-ligation reaction to make the library of all single and double mutants (*Engler and Marillonnet, 2013*).

### The bacterial two-hybrid assay

The bacterial two-hybrid assay is based on the triple-plasmid system reported in Raman et al. (*Raman et al., 2016*) (*Figure 1—figure supplement 1*). The PDZ variants are expressed as fusions with the λcI DNA-binding domain under control of a *lac* promoter (in PZS22 plasmid, low-copy SC101 origin, trimethioprim (Tm) resistance), the PDZ ligand is expressed as a fusion with the RNA polymerase α-subunit under control of a *tet* promoter (in PZA31 plasmid, low-copy p15A origin, kanamycin (Kan) resistance), and the reporter gene is *cat* (coding for the enzyme chloramphenicol acetyltransferase), encoded by the pZE1RM plasmid (medium-copy number ColE1 origin and ampicillin (Amp) resistance).

Libraries of PDZ variants are transformed into electrocompetent pZE1RM$^+$pZA31$^+$ MC4100Z1 cells that harbor chromosomal copies of the *lac* repressor lacIq and the *tet* repressor TetR (*Tan et al., 2009*). After recovery in SOC medium, the culture is used to inoculate 50 mL of LB media (1:50 dilution) supplemented with 50 μg/mL Amp, 40 μg/mL Kan, and 20 μg/mL Tm and incubated overnight at 37°C with shaking. After ~12 hr, a 1:1000 dilution of the culture is made into fresh LB media with the three antibiotics and allowed to grow at 37°C to bring the cells into exponential growth ($OD_{600}$ = 0.1), at which point another 1:100 dilution is made into LB supplemented with the three antibiotics and 50 ng/mL of doxycycline hydrochloride (dox) to induce expression of the PDZ ligand fusion. Cells are incubated at 25°C for 2 log-orders of growth (~6.7 doublings) to allow protein expression to reach steady-state (*Poelwijk et al., 2011*). Growth at 25°C appears to represent an optimum in maximizing dynamic range whole also focusing assay sensitivity to binding affinity rather than protein stability. After induction, cells are 1:100 diluted into fresh LB media supplemented with all antibiotics and dox, and also chloramphenicol at a final concentration of 150 μg/mL for selection. Cells are grown at 25°C and harvested at $OD_{600}$ = 0.1, and plasmids are purified. The region covering the α2 helix is PCR amplified and Truseq barcodes and sequencing adapters (Illumina) are appended in two sequential PCR reactions. Truseq barcodes permit multiplexing different experiments in a single sequencing run.

### Deep sequencing

To analyze allele distributions, samples are combined and sequenced on either the Illumina Miseq or Hiseq2500 instruments (University of Texas Southwestern Medical Center Genomics and Microarray Core) and subsequently de-multiplexed, with allele counts extracted from sequencing files using

FASTX-Toolkit (http://hannonlab.cshl.edu/fastx_toolkit/) and BioPython (*Cock et al., 2009*) and converted to frequencies before and after selection in Matlab or Python.

To relate sequencing data to binding free energies, we compute the relative frequency of each allele $x$ after ($s$) and before ($u$) selection: $z^x = f_s^x / f_u^x$. Since selection has the property that frequencies of alleles will exponentially diverge as a function of time, we take the logarithm after normalizing $z^x$ over all alleles to define the 'relative enrichment': $\Delta E^x = \ln(z^x / \sum_i z^i)$. Since selection in the BTH is designed to be proportional to the fraction bound of each PDZ variant to the target ligand (*McLaughlin et al., 2012*; *Raman et al., 2016*), we have that $\Delta E^x = a \ln(f_b^x) + C$ (Equation 2), where in the pseudo first-order limit $f_b^x = L / (L + K_d^x)$. $L$ (the free ligand concentration in vivo), $a$, and $C$ are free parameters determined by fitting $\Delta E^x$ measured for a library of 45 mutants of PSD95 with known equilibrium binding constants (Fig. S1B). This 'standard curve' for the BTH shows excellent dynamic range in reporting binding free energies over the full range of values for essentially all mutants in all homologs (*Figure 1D–E*). To determine binding energies for data acquired in different sequencing experiments, we use the known binding affinity of the wild-type alleles to apply a correction to $\Delta E^x$ scores to match the values determined for the standard curve experiment. Given the fitted parameters and a set of corrected $\Delta E^x$ scores, we compute equilibrium binding constants and corresponding free energies using Equation 2.

## Data analysis and statistical comparisons

Distributions of thermodynamic coupling energies for each pair of positions in each homolog were plotted and fitted to single or double Gaussian models using the Gaussian mixture modeling tools in MATLAB (Mathworks Inc.). For each position pair, the choice between these two models was made by selecting the model with the minimum Bayes Information Criterion (BIC), which appropriately penalizes more complex models in accounting for the data. For correlation analyses comparing the experimental data with coevolution-based predictions (*Figures 6* and *8*), linear models were fit in MATLAB. The significance of the fitted Pearson's coefficient of determination ($r^2$) is given by the F-test, with the null hypothesis that there is no correlation.

## Statistical coupling analysis (SCA)

SCA was performed using pySCA 6.0 as recently described (*Rivoire et al., 2016*) using two manually adjusted, structure-based alignments of 240 ('Lockless') or 1689 ('Poole') eukaryotic PDZ domains, or using a publicly available alignment from PFAM (9610 seqs, [*Finn et al., 2016*]). Briefly, SCA involves computing a conservation weighted correlation matrix $\tilde{C}_{ij}^{ab} = \phi_i^a \phi_j^b (f_{ij}^{ab} - f_i^a f_j^b)$, where $f_i^a$ and $f_{ij}^{ab}$ represent the frequency and joint frequencies of amino acids $a$ and $b$ at positions $i$ and $j$, respectively, and $\phi = \frac{\partial D}{\partial f} = ln\left[\frac{f(1-q)}{q(1-f)}\right]$, the gradient of the Kullback-Leibler entropy $D$ describing the degree of conservation of amino acids (*Halabi et al., 2009*). $\tilde{C}_{ij}^{ab}$ is reduced to a positional coevolution matrix $\tilde{C}_{ij}$ by taking the Frobenius norm over amino acid pairs for each $(ij)$, and $\tilde{C}_{ij}$ is subject to eigenvalue decomposition. Coevolving positions are hierarchically organized into one or more collective modes (protein sectors, [*Halabi et al., 2009*]) in the top eigenmodes of the SCA positional coevolution matrix, with lower modes indistinguishable from noise due to limited sampling (*Halabi et al., 2009*). For PDZ, a protein with a single hierarchical sector, we consider here just the top eigenmode, permitting calculation of cleaned coevolution matrix $\hat{C} = v_1 \lambda_1 v_1^T$, where $\lambda_1$ is the top eigenvalue and $v_1$ is the first eigenvector. The portion of $\hat{C}$ corresponding to the α2 helix is shown in *Figure 4D*.

## Direct coupling analysis (DCA)

DCA calculations were carried out for alignments of 1689 ('Poole'), 9610 (PFAM), or 102410 ('Hopf', [*Hopf et al., 2017*]) PDZ domains using the pseudolikelihood maximization approach reported in (*Hopf et al., 2017*), resulting in intrinsic constraints ($h_i$) for each amino acid and pairwise couplings ($J_{ij}$) for each amino acid pair at positions $i$ and $j$. These parameters define a statistical energy for any given amino acid sequence $\sigma = (\sigma_1 \ldots \sigma_L)$: $E(\sigma) = \sum_i h_i \sigma_i + \sum_{i<j} J_{ij} \sigma_i \sigma_j$. As described (*Hopf et al., 2017*), we use these parameters to compute the energetic effect of single mutants and pairwise

coupling of mutation pairs (*Equation 1*, main text), starting from the sequences of each homolog. Just as for the experimental data, histograms of all amino acid couplings for every pair of positions were fit to either single or double Gaussian distributions, and mean values used for comparisons with the experimental data for individual homologs (*Figure 8—figure supplement 2*). For homolog averaged couplings (*Figure 4G*), coupling energies for each amino acid pair were averaged over homologs, and then used to make histograms. For *Figure 4H*, we defined a DCA model in which top couplings in $J_{ij}$ were defined using a cutoff as in (*Ovchinnikov et al., 2014*), and all couplings below the cutoff were randomly scrambled. The resulting model parameters were used for computing the energetic effect of single and double mutations, as above.

## Acknowledgements

We thank FJ Poelwijk, MA Stiffler, C Nizak, O Rivoire, M Weigt, R Monasson, and S Cocco for discussions and technical advice, and members of the Ranganathan laboratory for critical review of the manuscript. We also thank the High Performance Computing Group (BioHPC) and the Genomics Core at UT Southwestern for providing computational resources and sequencing, respectively. This work was supported by NIH Grant RO1GM12345 (to RR), a Robert A Welch Foundation Grant I-1366 (to RR), and the Green Center for Systems Biology at UT Southwestern Medical Center. VS was supported in part through a pre-doctoral fellowship (NIGMS T32 GM008203).

## Additional information

### Funding

| Funder | Grant reference number | Author |
| --- | --- | --- |
| Welch Foundation | I-1366 | Rama Ranganathan |
| National Institutes of Health | RO1GM12345 | Rama Ranganathan |

The funders had no role in study design, data collection and interpretation, or the decision to submit the work for publication.

### Author contributions

Victor H Salinas, Conceptualization, Software, Formal analysis, Supervision, Funding acquisition, Validation, Visualization, Writing—review and editing; Rama Ranganathan, Conceptualization, Data curation, Software, Formal analysis, Validation, Investigation, Visualization, Methodology, Writing—original draft, Project administration, Writing—review and editing

### Author ORCIDs

Rama Ranganathan http://orcid.org/0000-0001-5463-8956

### Decision letter and Author response

Decision letter https://doi.org/10.7554/eLife.34300.028
Author response https://doi.org/10.7554/eLife.34300.029

## Additional files

### Supplementary files

• Supplementary file 1. Salinas_Ranganathan_data.
DOI: https://doi.org/10.7554/eLife.34300.023

• Transparent reporting form
DOI: https://doi.org/10.7554/eLife.34300.024

### Data availability

Mutation data have been deposited in the Dryad database under accession code doi:10.5061/dryad.gk4m1

The following dataset was generated:

| Author(s) | Year | Dataset title | Dataset URL | Database, license, and accessibility information |
|---|---|---|---|---|
| Salinas VH, Ranganathan R | 2017 | Data from: Inferring amino acid interactions underlying protein function | http://dx.doi.org/10.5061/dryad.gk4m1 | Available at Dryad Digital Repository under a CC0 Public Domain Dedication |

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
