## [Decision Letter]

Thank you for submitting your article "Inferring amino acid interactions underlying protein function" for consideration by *eLife*. Your article has been reviewed by three peer reviewers, evaluation has been overseen by a Reviewing Editor and Detlef Weigel as the Senior Editor. The following individual involved in review of your submission has agreed to reveal their identity: Nir Ben-Tal (Reviewer #1); Amnon Horovitz (Reviewer #2).

The reviewers have discussed the reviews with one another and the Reviewing Editor has drafted this decision to help you prepare a revised submission.

Summary:

The manuscript presents deep sequencing study of the ligand-binding helix of 5 homologous PDZ domains. The data is analyzed, using double-mutant thermodynamics couplings, to estimate energetic couplings between the 9 amino acid positions of the helix. The results are correlated with calculated evolutionary couplings using both the SCA method, developed by the authors, and the DCA method used to infer direct couplings. The conclusion is that while DCA detects direct couplings, useful within the context of structure prediction, SCA appears to detect 'functional', allosteric couplings.

Opinion:

Assuming that it was done well, the deep sequencing data itself, regardless of the analysis and interpretations, can be very handy and useful for other studies. Its interpretation using the double-mutant cycle is insightful (but raises several open questions), and the comparison to the evolutionary couplings is questionable.

Essential revisions:

The energetic couplings:

1) A single Gaussian appears to fit well to many (perhaps most) of the energetic couplings in Figure 2, but the description in the main text (subsection “A deep coupling scan in the PDZ family”, second paragraph) refers to double-Gaussian. The energetic couplings of the rest of the PDZ domains (S3-S6) has even fewer double-Gaussians. Thus, the description is confusing.

2) The proposed explanation for the double-Gaussian are two conformations, which is surprising for a helix. Which conformations would these be? The model in S8 is too abstract and does not explain.

3) The analysis considers all the mutation types to be equal but that's not the case. Mutations to glycine or proline, for example, are more likely to disrupt the helix structure whereas mutations to alanine are more likely to be non-disruptive. Given that a helix was studied here, special consideration should be given to helix disrupting mutations. Some re-analysis according to mutation types and discussion of this issue is needed. Perhaps, the bimodal distributions (double-Gaussians) are due to mutation types?

4) Given that this study focuses on an α-helix, certain couplings (both energetic and evolutionary) such as between i,i+3,4 are expected. These couplings may have little to do with the PDZ fold and simply reflect helix properties and solvent exposure. A previous analysis of correlated mutations in the helices of many unrelated proteins did indeed reveal enrichment at positions i,i+3,4 (Noivirt et al. PEDS 2005). Such couplings are detected essentially only along one face of the helix. On the one hand we wonder why it is missing in others. On the other hand, the observed coupling should be discussed in view of this anticipation. That is, maybe the 1-4 and 1-5 couplings (Figure 2) simply reflect an α-helix.

5) What is the mechanics of energy transduction between the amino acids? Which forces are involved? This is particularly interesting for amino acids that are not in direct contact with each other.

6) The mechanistic and biological implications of the couplings should be discussed further. Why is allostery needed in PDZ domains? Why is it not completely shared by all 5 homologues? How does the specific coupling observed for each homologue serve its unique function?

7) Why limiting the analysis to a single helix? It would have been much more insightful to cover the whole domain. Especially given that it's so small.

8) The deep sequencing data should be made publicly available and easily accessible.

Coevolution:

9) We have some concern about the correlations in Figure 5. To what extent are the differences meaningful given that most of the data points are clustered together and the differences between plots appear to be due to a few outliers?

10) With so many homologous sequences, it should be possible to examine the robustness of the results using k-fold tests. For example, how does evolutionary coupling computed using (randomly chosen) half of the taxa compare to the values obtained with the other half? And how do the couplings in each set correlate with the energetic couplings?

11) Regarding the previous point, it would be insightful to examine the robustness of the two methods used to estimate coevolution. Both DCA, used to detect direct couplings, and SCA, used to suggest 'functional couplings'. Hopefully both will be equally robust in k-fold tests.

12) "Extracting the coevolution pattern in the top eigenmode for just the α2 helix (Figure 5C), we find that coevolution as defined by SCA in fact nearly quantitatively recapitulates the homolog averaged experimental couplings collected here (𝑟^2^ = 0.82, 𝑝 = 10;<= by F-test, Figure 5D)." *r*^2^ = 0.82 is fair correlation at most. This sentence should be tuned down considerably.

13) The statement in the sentence before the Discussion that non-contact couplings in the DCA model represent noise is at odds with Anishchenko et al., 2017. This discrepancy requires some discussion.

14) When comparing the two coevolution methods the authors should make the most of each. For some reason they use a very small alignment in DCA. About one tenth of all possible sequences.

15) DCA (and a full evolutionary study) takes into account both direct and indirect couplings between all residues. SCA on the other hand, takes into account only couplings between amino acid pairs. Thus, the energetic couplings work, via the double-mutant cycle, is more similar in spirit to SCA. The authors should refer to this point when correlating the two evolutionary coupling methods to the energetic coupling analysis.

[Editors' note: further revisions were requested prior to acceptance, as described below.]

Thank you for resubmitting your work entitled "Coevolution-based inference of amino acid interactions underlying protein function" for further consideration at *eLife*. Your revised article has been favorably evaluated by Detlef Weigel (Senior Editor), a Reviewing Editor, and outside reviewer.

The manuscript has been improved but there are some remaining issues that need to be addressed before acceptance, as outlined below:

This paper is important in two regards. First, it describes a high-throughput approach to mutant cycle analyses. Second, it shows the relative strengths and weaknesses of DCA and SCA. Nevertheless, there still are some concerns after the revision. The main one is that the equation in Figure 4 seems wrong. This may be a typo but if not, then the analysis in the paper based on this equation is also wrong. The correct form should be:

K_x_^app^ = K_x_(1+K_c_*α)/(1 + K_c_)

Other comments:

1) Averaging over mutation types is better than some arbitrary choice but, under favorable circumstances, mutations to alanine are preferred as a reference because with this substitution interactions are mostly removed without new ones being introduced.

2) In previous work (Cell 2009), the authors attributed the top eigenmode to evolutionary noise but not in this paper. This needs explaining.

3) We still think that the coincidence of the pattern of couplings observed in this paper with i, i+3,4 periodicity in helices suggests that maybe they reflect secondary structure and not allostery.

4) It should be noted that correlated mutations between distant residues can also reflect negative design (Noivirt PLoS Comp. Biol. 2009).

---

## [Author Response]

Essential revisions:The energetic couplings:1) A single Gaussian appears to fit well to many (perhaps most) of the energetic couplings in Figure 2, but the description in the main text (subsection “A deep coupling scan in the PDZ family”, second paragraph) refers to double-Gaussian. The energetic couplings of the rest of the PDZ domains (S3-S6) has even fewer double-Gaussians. Thus, the description is confusing.

In fitting distributions, model selection was carried out by statistical criteria, following the Bayes Information Criterion (BIC), which correctly penalizes models with more parameters and provides a rigorous basis to choose single or double Gaussian models. We have added a section in the Materials and methods to describe the fitting process.

2) The proposed explanation for the double-Gaussian are two conformations, which is surprising for a helix. Which conformations would these be? The model in S8 is too abstract and does not explain.

We agree that the description of the model was too cursory in the text and buried in the supplementary information. In revision, we have now moved the model to a main figure and have added a main text section, which provides clearer explanations and connections to known biological properties of PDZ domains (subsection “A model for distributions of thermodynamic mutant cycle couplings”). We also have added a paragraph in the Discussion section to establish the connections between the model and the main conclusions of the paper (second paragraph).

But, it’s important to make a more general point. In agreement with the reviewers, a main conclusion from our data is that the two-state-like behavior of the internal conformational equilibrium is not well described by known average chemical properties of secondary structure or amino acids. Instead, we show that the four positions in the helix that are engaged in the internal equilibrium are actually part of a globally correlated coevolving mode that spans the full protein structure, connecting several secondary structure elements and mediating allosteric communication in PDZ domains (new Figure 7). Thus, we propose that the unit of protein structure from which the two-state behavior emerges is the collective mode, not any particular structural element. This re-formulation of basic functional units of proteins from secondary structure elements to global collective modes is one important feature of these data, and we have tried in revision to make this key point clearer.

3) The analysis considers all the mutation types to be equal but that's not the case. Mutations to glycine or proline, for example, are more likely to disrupt the helix structure whereas mutations to alanine are more likely to be non-disruptive. Given that a helix was studied here, special consideration should be given to helix disrupting mutations. Some re-analysis according to mutation types and discussion of this issue is needed. Perhaps, the bimodal distributions (double-Gaussians) are due to mutation types?

We fully sympathize with the reviewers’ intuition about the character of mutations, but in fact the data indicate that there is no simple relationship between mutation types or helical propensities and thermodynamic couplings due to mutations. In revision, we show this directly with new figures that plot the distribution of coupling energies for every one of the 20 amino acid substitutions (Figure 3—figure supplement 1A). We also show that mutations to Gly or Pro do not show obvious evidence for greater couplings with other mutations than do Ala mutations (Figure 3—figure supplement 1B-D). These findings may at first glance be surprising, but they simply reinforce

the sense that functional couplings seem to be more about the heterogeneous character of positions rather than the heterogeneous character of amino acids. We do not argue that mutations are all equal; instead the data support the argument that over the full ensemble of substitutions, mutations act as Gaussian perturbations around the native coupling between amino acids, with only the slight twist that for some positions, the perturbations influence a nonlinear internal equilibrium. Thus, the average value over a large ensemble of pairwise mutations provides the best estimate of the native coupling from mutation experiments. In revision, we have made these points more directly.

4) Given that this study focuses on an α-helix, certain couplings (both energetic and evolutionary) such as between i,i+3,4 are expected. These couplings may have little to do with the PDZ fold and simply reflect helix properties and solvent exposure. A previous analysis of correlated mutations in the helices of many unrelated proteins did indeed reveal enrichment at positions i,i+3,4 (Noivirt et al. PEDS 2005). Such couplings are detected essentially only along one face of the helix. On the one hand we wonder why it is missing in others. On the other hand, the observed coupling should be discussed in view of this anticipation. That is, maybe the 1-4 and 1-5 couplings (Figure 2) simply reflect an α-helix.

While some experimental couplings are consistent with the classic periodicity of hydrogen-bonding that defines an α-helix, others do not (e.g. 1-8. 4-5, and many specific couplings in individual domains). In addition, and as noted by the reviewers, there are many other i+3/i+4 couplings that are zero. As argued in the paper, the pattern of couplings is more consistent with perturbation of a collectively acting group of amino acids that emerge from the tertiary structure (Figure 7) than anything to do with the characteristics of the α2 helix as an isolated element. The reviewers are correct that our study is in the helix, but the helix is embedded in the full protein, and thus the pattern of energetic couplings must be thought of as a global property of the protein, not a local property of secondary structure. In revision, we make these points clearer.

5) What is the mechanics of energy transduction between the amino acids? Which forces are involved? This is particularly interesting for amino acids that are not in direct contact with each other.

Well, this is an excellent question, but our study here is about the pattern of thermodynamic couplings, and not at all about the underlying mechanism. Nevertheless, given the data, we can speculate that the dense network of thermodynamic couplings between positions 1, 4, 5, and 8 likely indicate that they act as a collective mechanical unit within an otherwise weakly coupled environment. Such an arrangement would selectively transmit forces within the unit, underlying the long-range couplings we observe here. Independent physical experiments are underway to test this model and directly expose the mechanical basis for long-range coupling.

6) The mechanistic and biological implications of the couplings should be discussed further. Why is allostery needed in PDZ domains? Why is it not completely shared by all 5 homologues? How does the specific coupling observed for each homologue serve its unique function?

We agree, and have now provided much better descriptions of the known biological roles of allosteric communication and regulation in PDZ domains. Indeed, there is excellent evidence for long-range allosteric control in PDZ domains that corresponds to the collective mode (new Figure 7), and for the involvement of the very residues in the α2 helix that we identify as experimentally coupled in this work.

7) Why limiting the analysis to a single helix? It would have been much more insightful to cover the whole domain. Especially given that it's so small.

There are two answers to this question. First, a full analysis of all double mutants in the PDZ domain would involve ~ 2 x 10^6^ mutations per protein, a scale that may be possible, but is hardly trivial while maintaining experimental quality and homolog averaging. This is beyond the reasonable scope now. But the second answer is more important. Regardless of any proposed experiment, all these so-called “deep mutational” studies ultimately lack scalability and generality; indeed, they are technically difficult to optimize, they are very expensive, and they will always be limited in scope. Thus, the primary value of the experimental data here is to provide benchmark data to evaluate other approaches that can give us real power in generalized analyses of epistatic interactions within proteins. In that regard, we respectfully suggest that the study we carry out here provides exactly the kind of high-quality experimental data over homologs that we need. The support for this statement is evident in (new) Figures 6 and 8, which show that conserved experimental couplings can indeed be predicted with high statistical significance from statistical genomic methods (Figure 6B-D). Based on these findings, the new Figure 7 shows the full prediction for the pattern of conserved couplings in the whole PDZ domain, an analysis that can now be done for many proteins.

8) The deep sequencing data should be made publicly available and easily accessible.

Our laboratory is fully committed to open sharing of data, and indeed, all the data will be made available to the scientific community. Indeed, we are using the Dryad database to distribute the full dataset.

Coevolution:9) We have some concern about the correlations in Figure 5. To what extent are the differences meaningful given that most of the data points are clustered together and the differences between plots appear to be due to a few outliers?

The Pearson’s coefficient of determination (conventionally called the 𝑟^"^) provides a measure of linear correlation between two variables, but does not give the *significance* of this correlation. Indeed, is the correlation different from no correlation, given scatter in the data and number of measurements? To address this, we report the significance of the correlation in the p-value of the F-test, with the null hypothesis that the correlation is zero. Thus, we answer in the manuscript precisely what the reviewers are asking…the meaningfulness of correlations and of differences in correlation over coevolution methods. We now include the F-test p-values in the relevant figure panels.

In revision, we have also made these arguments more explicit and have added further analyses to show how the significance of the correlation depends on various parameters – alignment size, composition, and mathematical details of the both the SCA and DCA methods (new Figures 6 and 8, and many edits in the main text). For three independent alignments, SCA shows significant correlation with the experimental data (𝑝 = 10^-14^𝑡𝑜 10^-12^ by F-test), while DCA shows weaker or no significant correlation (𝑝 =. 001 𝑡𝑜. 09), where the conventional 𝑝 = 0.05 or below is used as a threshold for significance. In revision, we also now explain the likely explanations for these differences in predictive power, which should stimulate the scientific community to unify and improve the statistical coevolution methods.

10) With so many homologous sequences, it should be possible to examine the robustness of the results using k-fold tests. For example, how does evolutionary coupling computed using (randomly chosen) half of the taxa compare to the values obtained with the other half? And how do the couplings in each set correlate with the energetic couplings?11) Regarding the previous point, it would be insightful to examine the robustness of the two methods used to estimate coevolution. Both DCA, used to detect direct couplings, and SCA, used to suggest 'functional couplings'. Hopefully both will be equally robust in k-fold tests.

We agree that it is valuable to report the robustness of the methods to sequencing sampling. In revision, we show that SCA is highly robust to both choice of alignments that vary in size and method of construction (new Figures 6B-D), and random sampling of the alignment (Figure 6—figure supplement 1), and we show that DCA is less robust to alignment choice (Figures 8C-E) and random samplings of the best-performing alignment (Figure 8—figure supplement 1). The results are not surprising; SCA focuses on conserved correlations and is therefore robust to size and errors in alignments, and DCA focuses on all correlations and is sensitive to alignment composition and errors.

12) "Extracting the coevolution pattern in the top eigenmode for just the α2 helix (Figure 5C), we find that coevolution as defined by SCA in fact nearly quantitatively recapitulates the homolog averaged experimental couplings collected here (𝑟^2^ = 0.82, 𝑝 = 10;<= by F-test, Figure 5D)." r^2^ = 0.82 is fair correlation at most. This sentence should be tuned down considerably.

As noted above, the significance of any correlation is quantitatively given in the F-test p-value, but there is no doubt that words used to express this result can be a matter of individual judgement. We have toned down the language used in this regard to be simply reflective of just what the numbers say.

13) The statement in the sentence before the Discussion that non-contact couplings in the DCA model represent noise is at odds with Anishchenko et al., 2017. This discrepancy requires some discussion.

This matter is confusing and requires some explanation and clarity. We thank the reviewers for pointing it out, especially the need to directly address the issue in the paper. Anishchenko et al. show that for the top L/2 DCA couplings in the 𝐽_12_ matrix (where 𝐿 is the length of the protein), cases of apparently non-contacting residues are often resolved as true contacts by one of three explanations: (a) they are contacts induced by oligomerization, (b) they are contacts in other conformational states of the same protein, or (c) they are contacts in homologous members of the family. In our work reported here, we are talking about *all* couplings in the 𝐽_12_ matrix (not just the top ones), most of which are truly not-contacting in the tertiary structure. Thus, there is no discrepancy with the specific findings of Anishchenko et al., except perhaps to point out that the weak non-contacting couplings in the 𝐽_12_ matrix are not noise. They do contribute to function prediction (compare new Figures 8C and 8F), even if they do not contribute to structure prediction. This does argue for a revision of the main conclusions of Anishchenko et al. that evolutionary constraints in protein sequences are strictly in direct contacts. In revision, we have significantly edited the main text (both Results and conclusions) to clarify these points and more importantly, we explicitly provide a hypothesis for what the next steps in refining the DCA-like approach might be.

14) When comparing the two coevolution methods the authors should make the most of each. For some reason they use a very small alignment in DCA. About one tenth of all possible sequences.

Consistent with the reviewers’ request, we now report data for three different alignments in the case of both SCA and DCA, designed to give both methods an optimal chance to explain the functional data. The analysis shows that SCA is (expectedly) robust to alignment size and composition (new Figure 6B-D), while DCA is (expectedly) not (Figure 8C-E). An interesting finding is that DCA has the best performance with the manually adjusted, structure-based alignment we originally reported (Figure 8C, 1656 seqs, 𝑟^2^ = 0.34, 𝑝 =.001), and shows much poorer performance with larger, publicly available alignments (Figure 8D-E, with 9610 and 102410 seqs, respectively; 𝑟^2^ = 0.11 − 0.05, 𝑝 =.02 −. 09).

15) DCA (and a full evolutionary study) takes into account both direct and indirect couplings between all residues. SCA on the other hand, takes into account only couplings between amino acid pairs. Thus, the energetic couplings work, via the double-mutant cycle, is more similar in spirit to SCA. The authors should refer to this point when correlating the two evolutionary coupling methods to the energetic coupling analysis.

These assertions require correction and clarification, and we thank the reviewers for giving us the opportunity to do so. DCA makes an energy model (a sequence “Hamiltonian”) that is defined by intrinsic and pairwise interactions of amino acids (new Figure 8B) and tries to account for all the observed frequencies and pairwise correlations of amino acids in the alignment with these parameters. SCA makes no explicit assumption of model, and mathematically decomposes a conservation-weighted correlation matrix to discover clusters of coevolving amino acids. Since pairwise correlations in an alignment could arise from higher order interactions (or couplings) between amino acids, SCA is implicitly open to models with higher than pairwise interactions in the Hamiltonian, while DCA (as currently formulated) is not. We now more explicitly spell out these ideas in the revised manuscript.

As for mutant cycles, the mathematical relationship of thermodynamic coupling to sequence coevolution is not trivial and is an active area of investigation (see Poelwijk et al. PLoS Comp. Biol. 12, e10004771). What we can say is that homolog-averaged thermodynamic couplings are analogous to couplings in the sequence Hamiltonian, and therefore the comparisons with experimental data directly test the energy function inferred by the DCA formalism. Indeed, the prediction of mutational effects using the sequence Hamiltonian is precisely what the DCA community is proposing now (e.g. Hopf et al., 2017). What is less obvious is the mathematical relationship of energetic couplings and the top modes of the SCA coevolution matrix. Nevertheless, the fact that these top modes predict energetic couplings well provides an important clue for now formally unifying SCA and DCA to produce models for proteins that accurately estimate both structural contacts and functional effects. We expect that the work presented here will contribute to such an outcome.

[Editors' note: further revisions were requested prior to acceptance, as described below.]

The manuscript has been improved […] Nevertheless, there still are some concerns after the revision. The main one is that the equation in Figure 4 seems wrong. This may be a typo but if not, then the analysis in the paper based on this equation is also wrong. The correct form should be:K_x_^app^ = K_x_(1+K_c_*α)/(1 + K_c_)

Well, the equation is not wrong – it just depends on the way that one writes the definitions of the equilibrium constants. In our analysis (Figure 4A), the reactions 𝐾*_x_* and 𝛼𝐾*_x_* are defined as dissociation constants (𝐾*_x_*= [0𝐹][𝐿]/[0𝐵] and 𝛼𝐾*_x_*= [1𝐹][𝐿]/[1𝐵]) and the internal configurational equilibrium is defined in the following way: 𝐾*_c_*= 0𝐹/1𝐹 and 𝛼𝐾*_c_*= 0𝐵/1𝐵. If one uses these definitions, the apparent binding affinity is given by:

Kxapp=Kx1+Kc1α+Kc

and the fraction of bound protein is given by 𝑓*_B_* = [𝐿]/([𝐿] + Kxapp). This is what is given in the paper and in Figure 4A. But, as long as one maintains detailed balance, one is free to define the equilibrium constants differently. In that case, the equation for Kxapp will look different, but no insights or conclusions will differ. For example, if one defines 𝐾*_x_* and 𝛼𝐾*_x_* as association constants (𝐾*_x_* = [0𝐵]/[0𝐹][𝐿] and 𝛼 𝐾*_x_*= [1𝐵]/[1𝐹][𝐿]) and the internal configurational equilibrium is defined in the opposite way: 𝐾*_c_*= 1𝐹/0𝐹 and 𝛼𝐾*_c_*= 1𝐵0𝐵, then the equation for Kxapp will be:

Kxapp=Kx1+αKc1+Kc,

where now the fraction of bound protein is given by 𝑓*_B_*= [𝐿]/[L]+1Kxapp, since the apparent dissociation constant is the inverse of the apparent association constant. This is the equation indicated by the reviewers. A quick check of the expected limits of the reaction equilibria at the extreme conditions of 𝐾*_c_*very large or very small will show that both these equations, with equilibrium constants defined in their own ways, give exactly the same results and insights.

The bottom line is that the equation and analysis in Figure 4 are correct. However, we thank the reviewers for raising our awareness to the importance of defining our conventions for equilibrium constants in deriving formulae. We have edited the figure legend of Figure 4 to make the definitions explicit.

Other comments:1) Averaging over mutation types is better than some arbitrary choice but, under favorable circumstances, mutations to alanine are preferred as a reference because with this substitution interactions are mostly removed without new ones being introduced.

We fully understand and sympathize with the long-held principle that if one were forced to make a single mutation to estimate the native coupling energy between two positions, alanine mutations are preferred. But, what our work shows with considerable data is that while alanine mutations might be preferred over other substitutions for single thermodynamic cycles, the best estimator of native couplings is nevertheless to average over all possible combinations of mutations at a pair of sites. This is justified by the nature of the empirical distributions and it sensibly eliminates the random vagaries of couplings estimated by specific pairs of mutations. We show that if one can manage to do it, the average over the ensemble is better experiment that any particular choice of individual mutations.

2) In previous work (Cell 2009), the authors attributed the top eigenmode to evolutionary noise but not in this paper. This needs explaining.

In Halabi et al., 2009, the main result was the finding of multiple near independent sectors. For this purpose, the top eigenmode is irrelevant since it contains the coherent correlations of all sequence positions. Thus, independent sectors were represented in the next few eigenmodes, namely two to four. While it is true that the top mode contains phylogenetic noise (as do all modes to some extent), it also clearly contains significant signal. Accordingly, in Halabi et al., we did not remove the top mode; we simply ignored it. In follow up papers (including a methods focused paper – Rivoire et al., 2016, we have explained our advances in understanding of how to handle the analysis of the SCA matrix in cases of single or multiple sectors. The best approach is use analytical methods such as independent component analysis to study the independence (or not) of the coevolution pattern. We reference this study explicitly in this work and since we do not report any new advance in SCA here, we hope that our citations will suffice to explain the current practice of the SCA method.

3) We still think that the coincidence of the pattern of couplings observed in this paper with i, i+3,4 periodicity in helices suggests that maybe they reflect secondary structure and not allostery.

Well, the pattern of experimental couplings shows an exact agreement with the known allosteric mode represented by the sector in PDZ domains (Figure 7) and shows partial agreement with helical periodicity with notable major exceptions (e.g. the large 1-8 coupling, and the fact that the majority of i, i+3/i+4 couplings are zero). We believe the interpretation of these data can be left to the scientific community with the publication of these data.

To be clear, we are not challenging the basic principles of secondary structure, but we strongly feel that the secondary structure centric view of energetic interactions that stabilize protein states has caused all of us to be less open that we should about alternative structural models that might have greater explanatory power. With the collection and publication of datasets such as these, we might begin to build a basis for rigorously assessing these views.

4) It should be noted that correlated mutations between distant residues can also reflect negative design (Noivirt PLoS Comp. Biol. 2009).

We have added a sentence in the subsection “Coevolution-based inference of functional couplings” to reflect this point and have added the suggested citation.